
**Impact of organic acids on chloride depletion of inland**
**transported sea spray aerosols**
Bojiang Su[1], Zeming Zhuo[1], Yuzhen Fu[2,3], Wei Sun[2,3], Ying Chen[1], Xubing Du[1],
Yuxiang Yang[2,3],Si Wu[1], Fugui Huang[4], Duohong Chen[5], Lei Li[1,*], Guohua Zhang[2,6],
Xinhui Bi[2,6], and Zhen Zhou[1]
[1] Guangdong Provincial Engineering Research Center for On-line Source Apportionment
System of Air Pollution, Institute of Mass Spectrometry and Atmospheric Environment, Jinan
University, Guangzhou 510632, PR China
[2] State Key Laboratory of Organic Geochemistry and Guangdong Key Laboratory of
Environmental Protection and Resources Utilization, Guangzhou Institute of Geochemistry,
Chinese Academy of Sciences, Guangzhou 510640, PR China
[3] University of Chinese Academy of Sciences, Beijing 100039, PR China
[4] Guangzhou Hexin Analytical Instrument Limited Company, Guangzhou 510530, PR China
[5] State Environmental Protection Key Laboratory of Regional Air Quality Monitoring,
Guangdong Environmental Monitoring Center, Guangzhou 510308, PR China
[6] Guangdong-Hong Kong-Macao Joint Laboratory for Environmental Pollution and Control,
Guangzhou Institute of Geochemistry, Chinese Academy of Sciences, Guangzhou 510640, PR
China
*Correspondence to: Lei Li (lileishdx@163.com)



## Highlights

1. Half of the sea salt aerosol (SSA) particles could be assigned as the biological origin.

2. Organic acids considerably contribute to chloride depletion of SSA particles.

3. Biological organic coatings may inhibit heterogeneous reactions of SSA particles.


**Abstract.** Heterogeneous reactions on sea spray aerosols (SSA) are the main pathway
to drive the circulation of chlorine, nitrogen, and sulfur in the atmosphere. The release
of Cl will significantly affect the physicochemical properties of SSA. However, the
impact of organic acids and mixing state on chloride depletion of SSA is still unclear.
Hence, the size and chemical composition of individual SSA particles during the East
Asian summer monsoon were investigated by a single particle aerosol mass
spectrometer (SPAMS). According to the chemical composition, SSA particles were
classified into SSA-Aged, SSA-Bio and SSA-Ca. In comparison to the aged Na-rich
SSA particles (SSA-Aged), some additional organic species related to biological origin
were observed in SSA-Bio, and each of two types accounts for approximately 50% of
total SSA particles. SSA-Ca may associated with organic shell of Na-rich SSA particles,
which only accounts for ~3%. Strongly positive correlations between Na and organic
acids (including formate, acetate, propionate, pyruvate, oxalate, malonate, succinate,
and glutarate) were observed for the SSA-Aged ($r^2 = 0.52$, $p < 0.01$) and SSA-Bio ($r^2 =$
0.61, $p < 0.01$), indicating the significance of organic acids in the chloride depletion
during inland transport. The contribution of these organic acids to the chloride depletion
is estimated to be up to 34%. Interestingly, the degree of chloride depletion is distinctly
different between SSA-Aged and SSA-Bio. It is most probably attributed to the
associated organic coating in the SSA-Bio particles, which inhibit the displacement
reactions between acids and chloride. As revealed from the mixing state of SSA-Bio,
Cl / Na ratio increases with increasing phosphate and organic nitrogen, which is known
to originate from biological activities. This finding provides some basis for the





improvement of modeling simulations in chlorine circulation and a comprehensive
understanding of the effects of organics on chloride depletion of SSA particles.
**Keywords:**
Sea spray aerosols; individual particles; chloride depletion; mixing state; organic acids.



## 1  Introduction

As one of the largest natural sources of aerosols, sea spray aerosols (SSA) have a global flux of 2000-10000 Tg yr$^{-1}$ (Gantt and Meskhidze, 2013) and global average distribution of 10.1 μg m$^{-2}$ (Ma et al., 2008). SSA are highly complex mixtures, and the chemical composition and mixing state of original SSA depends on the components of local seawater and the mechanisms of formation (Wang et al., 2017). While fresh SSA particles contain approximately 90% sodium chloride (NaCl) in mass, multiphase reactions considerably affect the chemical composition and mixing state, and subsequently, the physical and chemical properties of SSA. The multiphase reactions of SSA, as have been widely reported in field experiments and laboratory studies (Ault et al., 2014; Ghorai et al., 2014; Ryder et al., 2015; Trueblood et al., 2016; Bondy et al., 2017; Martin et al., 2017; Bertram et al., 2018), drive the circulation of elements (e.g., C, O, N, S, P, Cl) affecting tropospheric chemistry and global ecosystem (Finlayson-Pitts, 2003).

As one of the most important reactions, chloride depletion, as shown in R1, in SSA by interacting with acidic species modifies the physicochemical properties of SSA.

$$\text{HA}_{(g\ or\ aq)} + \text{NaCl}_{(aq\ or\ s)} \rightarrow \text{NaA}_{(aq\ or\ s)} + \text{HCl}_{(g\ or\ aq)} \tag{R1}$$

where NaCl represents the major component of SSA, and HA represents acidic species (e.g., HNO$_3$, H$_2$SO$_4$, and organic acids). Generally, inorganic acids are considered as the major contributors to chloride depletion in SSA (Dasgupta et al., 2007; Laskin et al., 2012; Chi et al., 2015), represented as:



$\quad\quad$ $HNO_{3\,(g\,or\,aq)} + NaCl_{(s\,or\,aq)} \rightarrow NaNO_{3\,(aq)} + HCl_{(g\,or\,aq)}$ $\quad\quad\quad$ (R2)
$\quad\quad$ $H_2SO_{4\,(g\,or\,aq)} + NaCl_{(s\,or\,aq)} \rightarrow Na_2SO_{4\,(aq)} + HCl_{(g\,or\,aq)}$ $\quad\quad\quad$ (R3)
$\quad\quad$ However, growing evidence indicates that inorganic acids cannot fully explain the
$\quad\quad$ chloride depletion (Laskin et al., 2012). It is therefore proposed that organic acids
$\quad\quad$ should be included to further explain the mechanism of chlorine depletion in aged SSA
$\quad\quad$ (Ault et al., 2013; Wang and Laskin, 2014; Wang et al., 2015). Similarly, the
$\quad\quad$ heterogeneous reactions on SSA with organic acids (R-COOH) can be described as:
$\quad\quad$ $R\text{-}COOH_{(g\,or\,aq)} + NaCl_{(s\,or\,aq)} \rightarrow R\text{-}COONa_{(aq)} + HCl_{(g\,or\,aq)}$ $\quad\quad\quad$ (R4)
$\quad\quad$ $NaNO_{3\,(aq)} + R\text{-}COOH_{(g\,or\,aq)} \rightarrow HNO_{3\,(g\,or\,aq)} + R\text{-}COONa_{(s\,or\,aq)}$ $\quad\quad\quad$ (R5)
$\quad\quad$ (*s*, solid; *aq*, aqueous; and *g*, gaseous)
$\quad\quad$ The physicochemical properties of SSA could be substantially altered through the
$\quad\quad$ production of organic salts (Trueblood et al., 2016; Bertram et al., 2018). With
$\quad\quad$ ubiquitous existence in the atmosphere, some typical organic acids, such as formic acid,
$\quad\quad$ acetic acid, and oxalic acid, may potentially contribute to chloride depletion. Mochida
$\quad\quad$ et al. (2003) suggested that organic acids (e.g., oxalic acid and succinic acid) should be
$\quad\quad$ considered in the models in order to predict chloride depletion accurately. Through
$\quad\quad$ investigating the interactions between the pure NaCl and carboxylic acids in the
$\quad\quad$ laboratory, Laskin et al. (2012) even suggested interactions between carboxylic acids
$\quad\quad$ and NaCl was the main contribution of chloride depletion. Consistently, Ghorai et al.
$\quad\quad$ (2014) observed that some dicarboxylic acids might cause obvious chloride depletion
$\quad\quad$ under a specific meteorological condition. These dicarboxylic acids (e.g., malonic acid
$\quad\quad$ and succinic acid) are ubiquitous in urban and marine aerosols (Kawamura and Bikkina,


2016), which may come from the polluted continental outflow and the open ocean
(Bikkina et al., 2015).

Given the potential contribution of organic acids to the chloride depletion, the

understanding of the relative contribution and the influencing factors is still unclear.
The investigation of factors that affect chloride depletion is indispensable to understand
the ageing process of SSA and the ability to serve as cloud condensation nuclei (CCN)
(Drozd et al., 2014; Wang et al., 2015). The morphology of fresh SSA particles is core-
shell structure, which consists of salt-core dominated by sodium chloride and outer shell
covered by $K^+$, $Ca^{2+}$, $Mg^{2+}$, $SO_4^{2-}$, $Cl^-$ and organic components (Laskin et al., 2012;
Collins et al., 2014; Chi et al., 2015). These organic components such as alkanes, fatty
acids, sugars, dicarboxylic acids and phosphate may emitted from phytoplankton and
bacteria in the sea surface microlayer (SSML) (Gaston et al., 2010; Bikkina et al., 2015;
Cochran et al., 2017a; Cochran et al., 2017b; Wang et al., 2017), and the chemical
composition and size distribution of SSA particles could be greatly changed by wave
breaking. In recent literature, organic coatings on the particles resulted from
atmospheric oxidation of hydrocarbons of biogenic and anthropogenic origin may
significantly regulate uptake of $N_2O_5$ (Folkers et al., 2003; Ryder et al., 2015).

In the present study, a single particle aerosol mass spectrometer (SPAMS) was

used to investigate the particle size, chemical composition and ageing degree of
individual SSA particles after long-range inland transport during the summer monsoon,
to reveal the relative contribution of organic acids on chloride depletion and the
influencing factors. The displacement reactions on the SSA particles with various types



of organic acids are considered, and the results suggest a significant impact of organic
acids on chloride depletion during inland transport. The influence of biogenic organics
in the chloride depletion was also discussed.

## 2 Materials and Methods

### 2.1 Field site description

The sampling site is located at Nanling national background station (Mt. Tianjing,

24°41′56″ N, 112°53′56″ E; 1690 m a.s.l.), which was approximately 350 km north of
the South China Sea and 200 km north of the Pearl River Delta (PRD) region. It is also
surrounded by a national park forest (273 km$^2$), where there were barely anthropogenic
pollutants. However, under the influence of the East Asian summer monsoon, the air
mass originated from the South China Sea might cross the PRD region to the sampling
site. As can be seen in *Supplement* Fig. S1, four major cluster back trajectories of air
masses originated from the South China Sea and Indo China Peninsula transported
across inland regions to the sampling site within 72 h. During the sampling period, the
average relative humidity was 87%, the average temperature was 26.3°C, the wind
direction was mainly southwesterly, and the average wind speed was 10 m s$^{-1}$. More
detailed information on the meteorological data can be found in *Supplement* Fig. S2.

### 2.2 Instrumentation

Individual particles were analyzed using a SPAMS (Hexin Analytical Instrument

Co., Ltd., China) from 11 May to 3 June 2018. The SPAMS was used to on-line measure


the size and chemical composition of individual particles. The design and principles of
SPAMS have been reported in detail previously (Li et al., 2011). Briefly, the aerosols
are drawn into the aerodynamics lens. Then the collimated particles beam through two
continuous laser beams (Nd: YAG laser, 532 nm) with a pace of 6 cm. The obtained
time of flight and velocity are corresponding to the vacuum aerodynamic diameter. The
velocity of an individual particle is applied to trigger the pulse laser (Nd: YAG laser,
266 nm). Subsequently, the generated ion fragments are detected by a bipolar time-of-
flight mass spectrometer. Standard polystyrene latex spheres of 0.2-2.2 μm were used
to calibrate vacuum aerodynamic particle sizes ($d_{va}$) of the measured individual
particles.

It is also noted that aerosols were sampled through two parallel inlets. The first

one is a ground-based counterflow virtual impactor (GCVI model 1205, Brechtel
Manufacturing Inc., USA), sampling the cloud residual particles, dried from cloud
droplets (with size larger than 8 μm) during cloud event (i.e., when the relative humidity
was higher than 95% and the visibility was lower than 3 km) (Zhang et al., 2017). The
other one is a $PM_{2.5}$ sampling inlet, delivering fine particles during cloud free periods.

**2.3  Classification of SSA**


The general characteristic peaks of SSA particles include $m/z$ 23 $[Na]^+$, 39 $[K]^+$,

46 $[Na_2]^+$, 62 $[Na_2O]^+$, 63 $[Na_2OH]^+$, 81 $[Na_2{}^{35}Cl]^+$ and 83 $[Na_2{}^{37}Cl]^+$ (Collins et al.,
2014; Arndt et al., 2017; Martin et al., 2017). There are some additional organic peaks
of biological origin (such as at $m/z$ 58 $[C_2H_5NHCH_2]^+$, 59 $[N(CH_3)_3]^+$, 74


$[(C_2H_5)_2NH_2]^+$, -26 $[CN]^-$, -42 $[CNO]^-$, -63 $[PO_2]^-$, -79 $[PO_3]^-$, etc.), besides sodium-
related peaks in SSA particles (Ault et al., 2014; Sultana et al., 2017b). In this study,
SSA particles were identified by the presence of peaks at $m/z$ 23, 46, 62, 63, 81 and 83,
which was coincident with the previous study at the same site reported by Lin et al.
(2019). In addition, these organic signals are also considerable for identification of SSA
particles, when the above sodium related peaks exist.
A total of ~2 million detected particles were clustered into several groups using an
Adaptive Resonance Theory neural network (ART-2a) (Song and Hopke, 1999) with a
vigilance factor of 0.75, a learning rate of 0.05 and a maximum of 20 iterations. Based
on the above representative peaks, ~50 000 SSA particles were identified. Three types
of SSA particles with distinct mass spectral characteristics were obtained, including
~25 000 SSA-Aged, ~25 000 SSA-Bio and 1 500 SSA-Ca, respectively. We note that
the mass spectral characteristics of SSA for the cloud residual particles ensemble those
for the cloud-free particles. And therefore, we focus on the influence of long-range
transport, rather than in-cloud process, on the modification of SSA.
## 3.    Results and discussion
**3.1 General characteristics of inland transported SSA particles**
Figure 1 provides the averaged positive and negative ion mass spectra of three
types of SSA particles. SSA-Aged is characterized by prominent ion signature for $m/z$
at 23 $[Na]^+$, 39 $[K]^+$, 40 $[Ca]^+$, 46 $[Na_2]^+$, 62 $[Na_2O]^+$ and 63 $[Na_2OH]^+$. Some
contributions from $m/z$ 24 $[Mg]^+$, 56 $[Fe]^+$ or 56 $[CaO]^+$, 81 $[Na_2{}^{35}Cl]^+$, 83 $[Na_2{}^{37}Cl]^+$,


108 $[Na_2NO_3]^+$ and 165 $[Na_2SO_4]^+$ were also observed. These mass spectral
characteristics are similar to those in previous literature (Hughes et al., 2000; Collins et
al., 2014; Sultana et al., 2017a). This type of SSA particles with a significant ion marker
of Na is typically represented as Na-rich sea salt particles. In the negative ion spectra,
abundant nitrate were observed due to the ratio of $m/z$ at -46 $[NO_2]^-$, -62 $[NO_3]^-$, and -
147 $[Na(NO_3)_2]^-$, in contrast to weak chlorine ion signal at $m/z$ -35 $[^{35}Cl]^-$ and -37 $[^{35}Cl]^-$,
indicating that the particles have undergone partial but not fully atmospheric ageing
(Hughes et al., 2000; Sultana et al., 2017b). Several peaks are assigned as organic acids,
such as formate at $m/z$ -45 $[HCO_2]^-$, acetate at $m/z$ -59 $[C_2H_3O_2]^-$, propionate at $m/z$ -73
$[C_2HO_3]^-$, pyruvate at $m/z$ -87 $[C_3H_3O_3]^-$, oxalate at $m/z$ -89 $[C_2HO_4]^-$, malonate at $m/z$
-103 $[C_3H_3O_4]^-$, succinate at $m/z$ -117 $[C_4H_5O_4]^-$, and glutarte at $m/z$ -131 $[C_5H_7O_4]^-$
(Lin et al., 2019), which may be related to algal activity in the SSML or conversion of
second organic aerosols (SOAs) in the atmosphere (O'Dowd et al., 2014).

Compared with SSA-Aged, the averaged ion spectra of SSA-Bio are more

complex. SSA-Bio had significant additional signals from biological organic matter
(i.e., organic nitrogen and phosphate), besides the general characteristics of the SSA-
Aged particles (Prather et al., 2013; Guasco et al., 2014). While these markers might
also be associated with dust (Zawadowicz et al., 2017), it is most likely attributed to
biological markers herein, since there is negligible ion marker (e.g., Al, Ti, Si) for dust.
Distinct characteristics of amines (58 $[C_2H_5NHCH_2]^+$, 59 $[N(CH_3)_3]^+$) were presented
in the positive spectra, which is similar to the results in a prior laboratory study (Sultana
et al., 2017a). Besides, the source of amines could also be influenced by the formation





of secondary species (such as animal husbandry and biomass burning) during transport
(Cheng et al., 2018). The organic nitrogen (i.e., -26 $[CN]^-$, -42 $[CNO]^-$) has been
assigned to the ionization of amino acids in previous studies (Abneesh et al., 2005;
Czerwieniec. et al., 2005). Phosphate peaks at $m/z$ -63 $[PO_2]^-$ and -79 $[PO_3]^-$ are likely
assigned as the ionization of components such as phospholipids in biological cells
(Fergenson, 2004; Collins et al., 2013; Cochran et al., 2017a; Cochran et al., 2017b;
Nguyen et al., 2017). It is noted that SSA-Bio should be regarded as the SSA population
influenced by biological activity (Prather et al., 2013). In addition, the peaks of $m/z$ 56
represents $[CaO]^+$ / $[KOH]^+$ or $[^{56}Fe]^+$. In contrast to SSA-Aged, those aforementioned
organic acids exhibited higher peak signal in SSA-Bio. Similar to SSA-Aged, inorganic
acids (-46 $[NO_2]^-$, -62 $[NO_3]^-$ and -97 $[HSO_4]^-$) with strong ion signals were also
observed. Despite of the different mass spectral pattern, the behavior and inland
transport of SSA-Aged and SSA-Bio may be similar. As can be seen in Fig. S1, the
relative proportions of them keep stable in the different air masses. They also exhibit
similar size distribution, concentrating in size range of 0.4-0.7 μm and peaking around
0.5 μm (Fig. S3).

SSA-Ca is identified by relatively higher contributions from calcium-related

compounds at $m/z$ 40 $[Ca]^+$, 56 $[CaO]^+$, 57 $[CaOH]^+$, 75 $[Ca^{35}Cl_2]^+$, 77 $[Ca^{37}Cl_2]^+$, and
113 $[(CaO)_2H]^+$, whereas associated with smaller sodium peak than other types. The
negative spectra are dominated by nitrate, sulfate, organic nitrogen, phosphate and
chloride. This SSA population has been previously classified as "organic-carbon-
dominated (OC)" (Prather et al., 2013; Collins et al., 2014), likely resulted from the



coating of Na-rich particles through crystallization and precipitation (Sultana et al.,
2017a; May et al., 2018). The mass spectral characteristics of the Ca-rich SSA particles
are quite similar to those of lake spray aerosols (Axson et al., 2016; May et al., 2018).
However, SSA-Ca only accounts for a negligible fraction (3.2%) and thus will not be
covered in the following discussions.
**3.2 Contribution of organic acids to the chloride depletion in the SSA particles**

The linear correlations based on peak area between Na and chloride, sulfate, nitrate,

and organic acids in the SSA-Aged and SSA-Bio particles are shown in Fig. 2. As
expected, there are strong correlations between Na and nitrate in both the SSA-Aged
($r^2 = 0.79$, $p < 0.01$) and SSA-Bio ($r^2 = 0.86$, $p < 0.01$) particles. In addition, more than
99% of SSA particles are internally mixed with nitrate (Fig. 3). This indicates that
chemistry in Reaction 2 (R2) is prevalent during long-range transport (Bondy et al.,
2017). This is also consistent with previous studies regarding that nitric acid is a major
contributor to chloride depletion (Zhao and Gao, 2008; Chi et al., 2015; AzadiAghdam
et al., 2019). It is probably because the concentration of its precursor $NO_x$ (4.67 µg m$^-$
$^3$) suppresses those of other acids in the south China sector (Wang et al., 2016; Wu et
al., 2019).

Strong positive correlations between Na and organic acids are also observed in

both the SSA-Aged ($r^2 = 0.52$, $p < 0.01$) and SSA-Bio particles ($r^2 = 0.61$, $p < 0.01$).
Furthermore, very high number fractions (NFs) of organic acids are also found in SSA-
Aged (72%) and SSA-Bio (59%), as shown in Fig. 3. This indicates the possible





presence of organic salts in SSA particles and the substantial contribution of organic
acids to the chloride depletion. The detailed mixing state between SSA particles and
several detected organic acids, as shown in Fig. S4, indicates that formate, oxalate,
malonate, and glutarate are the dominant salts. The relative peak area (RPA) ratio (acids
/ (sulfate + nitrate + organic acids)) is further applied to roughly evaluate the relative
contribution of different acids (nitric acid, sulfuric acid, and organic acids) to the
chloride depletion of SSA particles (Fig. S5). In the ageing process of the SSA particles,
nitrate occupies a large proportion in the SSA-Aged (63-96%) and SSA-Bio particles
(64-95%), respectively. Notably, chloride depletion attributed to organic acids could
account for 2–34% in the SSA-Aged particles and 2–29% in the SSA-Bio particles. The
relative contribution of organic acids to chloride depletion has been reported to be
higher than 30% at the eastern United States coast (Braun et al., 2017) and up to 40%
in Southeast Asia (AzadiAghdam et al., 2019). The contribution of sulfuric acids (0-
10% versus 0-18 %) is the lowest, although it shows positive correlation ($r^2 = 0.24$, $p <$
0.01 versus $r^2 = 0.54$, $p < 0.01$) for the SSA-Aged and SSA-Bio particles, respectively
(Fig. 2). In addition, similar variations in peak areas of sulfate, nitrate and organic acids
were observed in the SSA-Aged and SSA-Bio particles throughout the sampling period
(Fig. S6), indicating a close connection of the formation mechanism between inorganic
and organic acids.
**3.3 Effect of particle type on chloride depletion**

Cl / Na value is typically applied to evaluate the ageing degree of SSA particles


(Laskin et al., 2012; Bondy et al., 2017). There is a significant difference of Cl / Na
between the SSA-Aged (1.9%) and SSA-Bio (5.4%) particles (Fig. 4). This result
reflects less chloride remaining in the SSA-Aged, attributed to more severe ageing. It
might also be supported by relatively weak positive correlation ($r^2 = 0.46$, $p < 0.01$)
between Na and Cl (Fig. 2). This result may be explained by the influence of chemical
composition and mixing state on the evolution of the SSA particles (Collins et al., 2014;
Quinn et al., 2015; Sultana et al., 2017b). Additionally, concentration calculation was
also further quantified the chloride depletion percentage using the following equation:

272        %Cl⁻ depletion = $(1.81 \times [Na^+] - [Cl^-]) / (1.81 \times [Na^+]) \times 100\%$        (R6)

where $[Na^+]$ and $[Cl^-]$ are mass concentrations (µg m⁻³), and 1.81 is the typical

mass ratio of Cl / Na in seawater (Zhao and Gao, 2008; Braun et al., 2017;
AzadiAghdam et al., 2019). The overall %Cl⁻ for the total SSA particles varying from
55% to 99% with an average of 78% (detailed in *Supplement* Table S1), which is similar
to the previous filed study in the PRD region reported by Chen et al. (2016). However,
SSA particles are not the only source of chlorine ion in the atmosphere (Lightowlers et
al., 1988). The excess $[Cl]^-$ produced by fuel combustion could lower the %Cl⁻, which
might explain the weak difference between the two assessment methods of ageing
degree of SSA particles. Hence, both assessment methods could effectively evaluate
the chloride depletion of SSA particles supported by a positive correlation ($r^2 = 0.47$, $p$
$< 0.01$) (Fig. S7).

Further analysis indicates that organic matter of biological origin might play an

important role in such inhibition of chloride depletion in the SSA-Bio particles. It is


supported by the relationship between peak area ratio of Cl / Na and the biological
origin markers (-26 [CN]⁻, -42 [CNO]⁻, -63 [PO$_2$]⁻ and -79 [PO$_3$]⁻) described in section
3.1. As shown in Fig. 5, peak area ratio of Cl / Na exhibits an increasing trend with both
phosphate (-63 [PO$_2$]⁻, -79 [PO$_3$]⁻) and organic nitrogen (-26 [CN]⁻ and -42 [CNO]⁻).
This direct evidence indicated phosphate might have a considerable effect on chloride
depletion in SSA particles. The relationship between Cl / Na and organic nitrogen is
also consistent with that reported in our previous filed observations at the same site (Lin
et al., 2019). Previous laboratory study results have also shown that reactivity could be
inhibited by the organic matter of biological origin (Ault et al., 2014). As shown in Fig.
S8, transmission electron microscopy (TEM) images clearly show NaCl core and
organic coating of the SSA particles with various thicknesses. The thicker organic
coating may inhibit the reactive uptake of HNO$_{3\ (g)}$ or N$_2$O$_{5\ (g)}$ to SSA particles (Folkers
et al., 2003; Ryder et al., 2014; Ryder et al., 2015), resulting in a less released Cl to the
atmosphere. Such organic coatings are mostly composed of long-chain hydrocarbon,
saccharides, carbohydrate, amine and anionic surfactant (Jayarathne et al., 2016;
Bertram et al., 2018), and thus have stronger hydrophobicity and probably inhibit the
occurrence of Cl transport of convection and diffusion (Bondy et al., 2017).
**4. Conclusion and atmospheric implication**

We investigated the chloride depletion of SSA particles after long-range inland

transport in south China, during a monsoon season. The SSA particles still account for
~3% of the observed submicron particles and are extensively internally mixed with



various acids. While the contribution of nitric acid dominates over other acids to the
chloride depletion, our results suggest that the role of organic acids should not be
neglected. Up to 34% of chloride depletion could be explained by diverse organic acids.
Our results add to the growing body of evidence that carboxylic acid may play a
significant role in acid displacement reactions (Ma et al., 2013). Given the substantial
influence of organic acids on the hygroscopic properties of SSA (Ghorai et al., 2014),
such processes may affect CCN / IN activities and lifetime of SSA (Knopf et al., 2014),
and thus should be considered in models to predict the climate impact of SSA accurately.
Currently, the calculation model of organic acids (especially water-soluble organic
compounds) to chloride depletion is still limited (Laskin et al., 2012; Xu et al., 2013).
Peng et al. (2016) suggested organic salts produced by NaCl react with dicarboxylic
acids inhibit the volatilization of HCl that is resulting in less chloride depletion. Our
data may improve the understanding of chloride depletion responsible for mixing state
of diverse organic acids in the future study.
In addition, we stress that there is a SSA type (e.g., SSA-Bio) likely attributed to
the biogenic origin, exhibiting distinctly different chloride depletion, in comparison
with the commonly observed SSA-Aged type. Our data indicate that organic matter of
biological origin might play an essential role in such inhibition of chloride depletion in
the SSA-Bio particles. As previously reported, the presence of organic coatings on SSA
particles could effectively influence the heterogeneous reactivity of SSA particles
(Ryder et al., 2015; Bondy et al., 2017). Considering the considerable contribution
(~50%) of the SSA-Bio particles to the overall SSA, such information should be useful



to improve that model results for the climate impact of SSA.

## Author contributions

GHZ, LL, and XHB designed the research. BJS, GHZ, and LL analyzed the data,
and wrote the manuscript. YZF, XBD, YC, YXY and WS conducted sampling work
under the guidance of GHZ, LL, and XHB. DHC had an active role in supporting the
sampling work. YZF performed the laboratory analysis of individual particles by
TEM/EDS. All authors contributed to the discussions of the results and refinement of
the manuscript.
**Data availability.** Data are available on request from Lei Li (lileishdx@163.com).

## Acknowledgement

This work was supported by the National Nature Science Foundation of China (No.
41905106), Guangdong International Science and Technology Cooperation Project
(2018A050506020) and Guangdong Foundation for Program of Science and
Technology Research (Grant No. 2019B121205006).
**Competing interests.** The authors declare that they have no conflict of interest.

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

Contributions of Sea Spray and Lake Spray Aerosol to Inland Particulate

Matter,     Environ.     Sci.     Technol.,     5,     405-412,

http://doi.org/10.1021/acs.estlett.8b00254, 2018.

Mochida, M., Umemoto, N., Kawamura, K., and Uematsu, M.: Bimodal size

distribution of C2-C4 dicarboxylic acids in the marine aerosols, J. Geophys.

Res. Lett., 30, http://doi.org/10.1029/2003gl017451, 2003.

Nguyen, Q. T., Kjær, K. H., Kling, K. I., Boesen, T., and Bilde, M.: Impact of fatty acid

coating on the CCN activity of sea salt particles, Tellus. B., 69,

http://doi.org/10.1080/16000889.2017.1304064, 2017.

O'Dowd, C., Ceburnis, D., Ovadnevaite, J., Vaishya, A., Rinaldi, M., and Facchini, M.

C.: Do anthropogenic, continental or coastal aerosol sources impact on a

marine aerosol signature at Mace Head?, Atmos. Chem. Phys., 14, 10687-

10704, http://doi.org/10.5194/acp-14-10687-2014, 2014.

Peng, C., Jing, B., Guo, Y. C., Zhang, Y. H., and Ge, M. F.: Hygroscopic Behavior of

Multicomponent Aerosols Involving NaCl and Dicarboxylic Acids, J. Phys.

Chem. A, 120, 1029-1038, http://doi.org/10.1021/acs.jpca.5b09373, 2016.

Prather, K. A., Bertram, T. H., Grassian, V. H., Deane, G. B., Stokes, M. D., DeMott, P.

541         J., Aluwihare, L. I., Palenik, B. P., Azam, F., Seinfeld, J. H., Moffet, R. C.,

Molina, M. J., Cappa, C. D., Geiger, F. M., Roberts, G. C., Russell, L. M., Ault,

543         A. P., Baltrusaitis, J., Collins, D. B., Corrigan, C. E., Cuadra-Rodriguez, L. A.,

Ebben, C. J., Forestieri, S. D., Guasco, T. L., Hersey, S. P., Kim, M. J., Lambert,



W. F., Modini, R. L., Mui, W., Pedler, B. E., Ruppel, M. J., Ryder, O. S.,
Schoepp, N. G., Sullivan, R. C., and Zhao, D. F.: Bringing the ocean into the
laboratory to probe the chemical complexity of sea spray aerosol, P. Natl. Acad.
Sci. USA., 110, 7550-7555, http://doi.org/10.1073/pnas.1300262110, 2013.
Quinn, P. K., Collins, D. B., Grassian, V. H., Prather, K. A., and Bates, T. S.: Chemistry
and related properties of freshly emitted sea spray aerosol, Chem. Rev., 115,
4383-4399, http://doi.org/10.1021/cr500713g, 2015.
Ryder, O. S., Ault, A. P., Cahill, J. F., Guasco, T. L., Riedel, T. P., Cuadra-Rodriguez, L.
A., Gaston, C. J., Fitzgerald, E., Lee, C., Prather, K. A., and Bertram, T. H.: On
the Role of Particle Inorganic Mixing State in the Reactive Uptake of N2O5 to
Ambient Aerosol Particles, Environ. Sci. Technol., 48, 1618-1627,
http://doi.org/10.1021/es4042622, 2014.
Ryder, O. S., Campbell, N. R., Morris, H., Forestieri, S., Ruppel, M. J., Cappa, C.,
Tivanski, A., Prather, K., and Bertram, T. H.: Role of organic coatings in
regulating n2o5 reactive uptake to sea spray aerosol, J. Phys. Chem. A, 119,
11683-11692, http://doi.org/10.1021/acs.jpca.5b08892, 2015.
Song, X., and Hopke, P. K.: Classification of single particles analyzed by ATOFMS
using an artificial neural network, ART-2A, Anal. Chem., 71, 860-865,
http://doi.org/10.1021/ac9809682, 1999.
Sultana, C. M., Al-Mashat, H., and Prather, K. A.: Expanding Single Particle Mass
Spectrometer Analyses for the Identification of Microbe Signatures in Sea
Spray        Aerosol,        Anal.        Chem.,        89,        10162-10170,



http://doi.org/10.1021/acs.analchem.7b00933, 2017a.

Sultana, C. M., Collins, D. B., and Prather, K. A.: Effect of Structural Heterogeneity in

Chemical Composition on Online Single-Particle Mass Spectrometry Analysis

of Sea Spray Aerosol Particles, Environ. Sci. Technol., 51, 3660-3668,

http://doi.org/10.1021/acs.est.6b06399, 2017b.

Trueblood, J. V., Estillore, A. D., Lee, C., Dowling, J. A., Prather, K. A., and Grassian,

573        V. H.: Heterogeneous Chemistry of Lipopolysaccharides with Gas-Phase

Nitric Acid: Reactive Sites and Reaction Pathways, J. Phys. Chem. A., 120,

6444-6450, http://doi.org/10.1021/acs.jpca.6b07023, 2016.

Wang, B., and Laskin, A.: Reactions between water-soluble organic acids and nitrates

in atmospheric aerosols: Recycling of nitric acid and formation of organic salts,

578        J. Geophys. Res. Atmos., 119, 3335-3351,

http://doi.org/10.1002/2013JD021169, 2014.

Wang, B., O'Brien, R. E., Kelly, S. T., Shilling, J. E., Moffet, R. C., Gilles, M. K., and

Laskin, A.: Reactivity of liquid and semisolid secondary organic carbon with

chloride and nitrate in atmospheric aerosols, J. Phys. Chem. A., 119, 4498-

4508, http://doi.org/10.1021/jp510336q, 2015.

Wang, N., Lyu, X. P., Deng, X. J., Guo, H., Deng, T., Li, Y., Yin, C. Q., Li, F., and Wang,

S. Q.: Assessment of regional air quality resulting from emission control in the

Pearl River Delta region, southern China, Sci. Total. Environ., 573, 1554-1565,

http://doi.org/10.1016/j.scitotenv.2016.09.013, 2016.

Wang, X. F., Deane, G. B., Moore, K. A., Ryder, O. S., Stokes, M. D., Beall, C. M.,





Collins, D. B., Santander, M. V., Burrows, S. M., Sultana, C. M., and Prather,

590       K. A.: The role of jet and film drops in controlling the mixing state of

submicron sea spray aerosol particles, P. Natl. Acad. Sci. USA., 114, 6978-

6983, http://doi.org/10.1073/pnas.1702420114, 2017.

Wu, Z., Zhang, Y., Zhang, L., Huang, M., Zhong, L., Chen, D., and Wang, X.: Trends

of outdoor air pollution and the impact on premature mortality in the Pearl

River Delta region of southern China during 2006-2015, Sci. Total. Environ.,

690, 248-260, http://doi.org/10.1016/j.scitotenv.2019.06.401, 2019.

Xu, G., Gao, Y., Lin, Q., Li, W., and Chen, L.: Characteristics of water-soluble inorganic

and organic ions in aerosols over the Southern Ocean and coastal East

Antarctica during austral summer, J. Geophys. Res. Atmos., 118, 13,303-

313,318, http://doi.org/10.1002/2013jd019496, 2013.

Zawadowicz, M. A., Froyd, K. D., Murphy, D. M., and Cziczo, D. J.: Improved

identification of primary biological aerosol particles using single-particle mass

spectrometry, Atmos. Chem. Phys., 17, 7193-7212, http://doi.org/10.5194/acp-

17-7193-2017, 2017.

Zhang, G., Lin, Q., Peng, L., Yang, Y., Fu, Y., Bi, X., Li, M., Chen, D., Chen, J., Cai,

Z., Wang, X., Peng, P., Sheng, G., and Zhou, Z.: Insight into the in-cloud

formation of oxalate based on in situ measurement by single particle mass

spectrometry, Atmos. Chem. Phys., 17, 13891–13901,

https://doi.org/10.5194/acp-17-13891-2017, 2017.

Zhao, Y., and Gao, Y.: Acidic species and chloride depletion in coarse aerosol particles



in the US east coast, Sci. Total. Environ., 407, 541-547,

http://doi.org/10.1016/j.scitotenv.2008.09.002, 2008.





# Figures

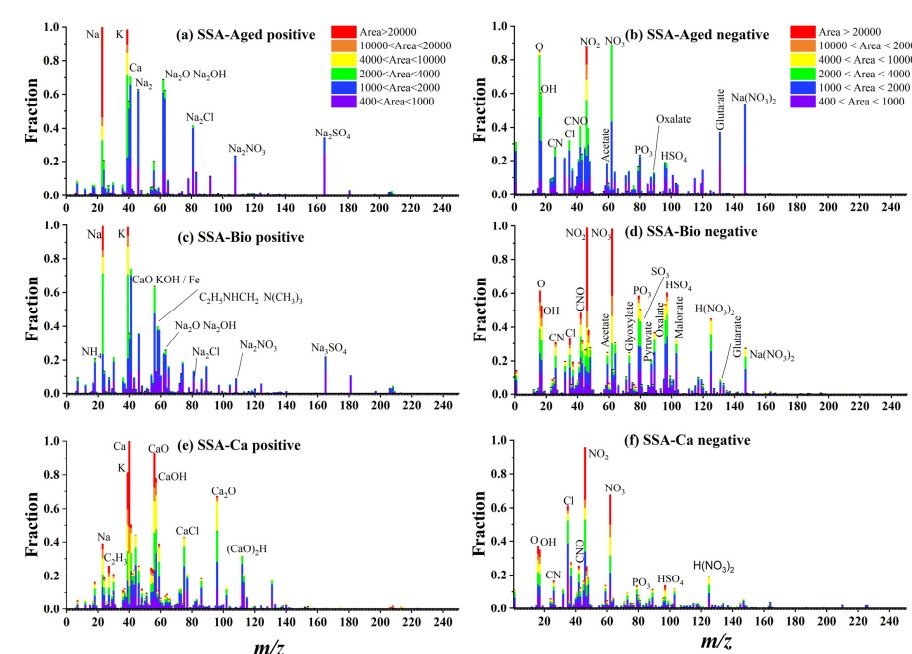

Figure 1. The averaged digitized positive and negative ion mass spectra of the major types of SSA particles.

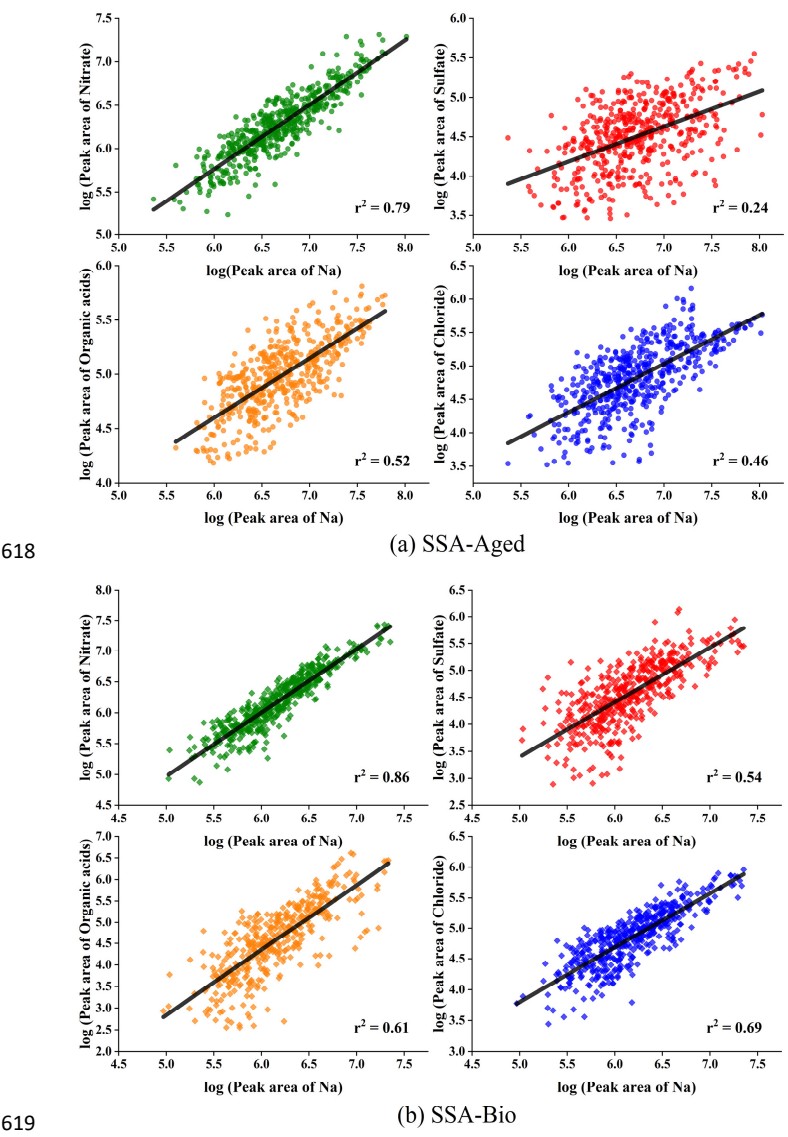

(a) SSA-Aged

(b) SSA-Bio

Figure 2. Correlations between hourly mean peak area of Na (*m/z* 23) and sulfate (*m/z*
-97), nitrate (*m/z* -46 and 62), organic acids (*m/z* -45, -59, -73, -87, -89, -103, -117 and
-131) and chloride (*m/z* -35 and -37) in the SSA-Aged and SSA-Bio. The data is
logarithmically transformed to follow a normal distribution.



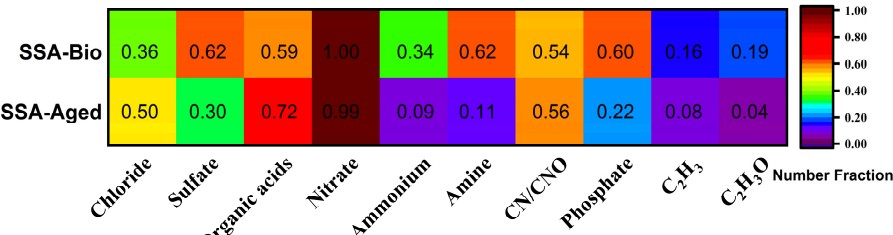

Figure 3. Hourly mean number fractions (NFs) of major component in the SSA-Aged

and SSA-Bio. The major component includes chloride (*m/z* -35 or -37), sulfate (*m/z* -

97), organic acids (*m/z* -45, -59, -73, -87, -89, -103, -117 or -131), nitrate (*m/z* -46 or -

62), ammonium (*m/z* 18), amine (*m/z* 58 and 59), organic nitrogen (*m/z* -26 or -42),

phosphate (*m/z* -63 or -79), and organic carbon (*m/z* 27 and 43).

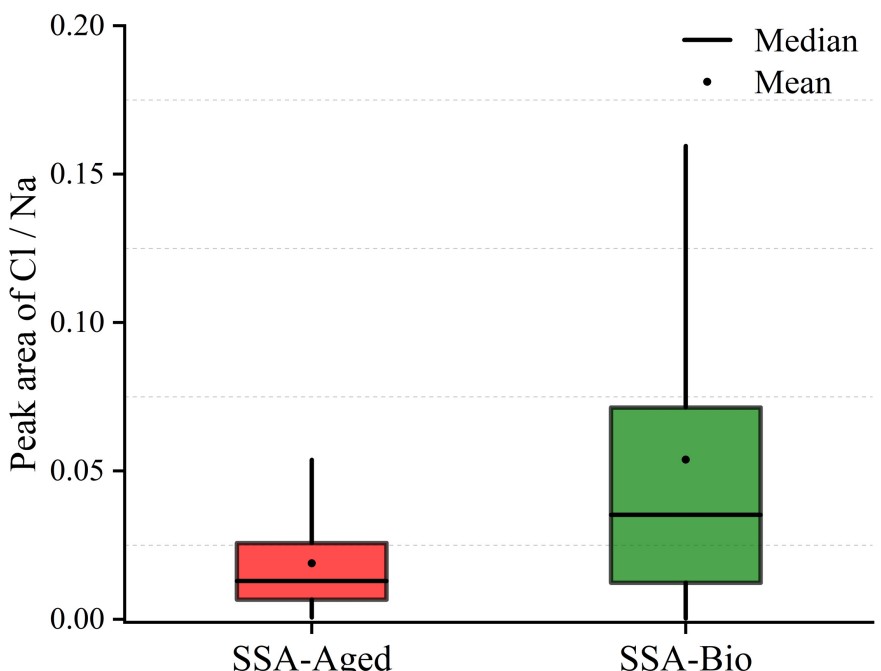


Figure 4. A box and whisker plot of hourly mean peak area of Cl / Na in SSA-Aged

and SSA-Bio. In the box and whisker plot, the lower and upper lines of the box denote
the 25 and 75 percentiles, respectively. The lower and upper edges denote the 10 and
90 percentiles, respectively.

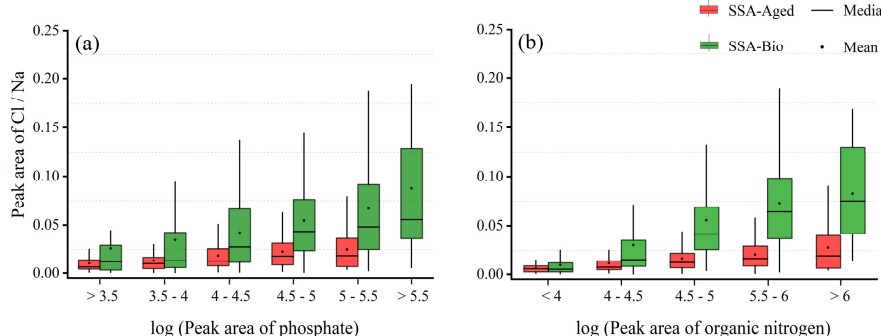


Figure 5. The logarithmical peak area of phosphate (*m/z* -63 and -79) and organic
nitrogen (*m/z* -26 and -42) varied as a function of hourly mean peak area ratio of Cl /
Na.