# Peer review of "Impact of organic acids on chloride depletion of inland"

_Atmospheric Chemistry and Physics, 2020_

## Referee Comment (RC1) · Anonymous Referee #1 · 7 Jul 2020

General comments

This paper discusses the depletion of chloride in sea spray aerosol (SSA) particles based on a field observation of atmospheric aerosol during the East Asian summer monsoon using a single particle aerosol mass spectrometer. The authors claim that up to 34% of chloride depression could be explained by organic acids. They also claim that SSA of biogenic origin exhibits chloride depression that is distinctly different from that of commonly observed aged SSA, and infer that organic coatings on SSA could influence on the heterogeneous reactivity of SSA particles. The subject of this study is important to understand the chemical processes of SSA and also their role in the atmosphere.

Although the findings from this study are potentially significant, the conclusions are

not fully supported by the experimental evidence provided in the current version of the manuscript. For example, in the quantitative analysis of the contributions of inorganic and organic acids on chloride depression, no explanation is provided about possible different sensitivity of the mass spectral analysis to different compounds. Further, the comparison of Cl/Na from mass spectra and that from mass concentrations was performed by omitting a substantial fraction of data points, without an adequate explanation on the rationale of the omission. To draw a solid conclusion from the data set presented in this paper, a thorough revision is necessary. More specific comments are listed below.

Specific comments

Page 3, lines 25-26: For nitrogen and sulfur, the explanation here sounds an overstatement. Page 6, line 80: Is there any experimental/theoretical evidence to support reaction R5? Page 9, lines 146-151: If both types of particles were subjected to the analysis, it should be explained explicitly. Page 10, line 163: What is the exact number of groups? Page 10, line 166: To justify the identification of SSA particles, a brief explanation on the differences between SSA particles and other types of particles should be provided. Page 13, line 222: What can crystallize? The meaning of "precipitation" is not clear. Page 13, lines 228-229: The definition of hourly-mean peak areas in Fig. 2 is unclear. Page 13, lines 232-233: This explanation is reasonable if nitrate was observed less frequently for non-SSA particles. Page 13, lines 236-238: It is difficult for readers to follow this part without an explanation about the mechanism of the suppression. Isn't it also possible that nitric acid and other acids have different reactivity to sea salts? Page 14, lines 245-246: The expression is misleading because the result in Fig. S4 is on number basis and may not relate to the mass concentrations of different organic salts in SSA particles. Page 14, lines 246-258: In the absence of an appropriate explanation on possible different sensitivities of this mass spectrometry to different types of chemical components (sulfate/nitrate/organics), the discussion here is questionable. Page 14, lines 258-261: Are the peak areas here hourly mean values?

[Figure]

Page 15, lines 264-265: Isn't there any possibility that matrix substances in the particles affect the relative sensitivity to detect Na and Cl? Page 15, lines 272-275: The mass concentrations of Na+ and Cl- seem to be from ion chromatography but there is no explanation about the methodology in the main manuscript. Page 15, line 280: Two assessment methods should be explained explicitly. Page 15, line 281-283: The analysis of the correlation is problematic because 7 among 22 set of data were omitted from the analysis without an appropriate assessment. Page 33, line 623: Which part of data shows a normal distribution? Why it is obtained by the logarithmic transformation? Page 35, lines 632-635: An explanation on the box and whisker plots should be given. Page 36, lines 637-639: Isn't it more reasonable to write Cl/Na as a function of peak areas of phosphate and organic nitrogen? Page 2, lines 7-11 (supplement): The association between trajectories on the map and C1-C4 should be explained. The graph superimposed on the map should also be explained. Page 4, lines 19-20 (supplement): If the detection efficiency for small particles was low, the peak shapes may be skewed. The possibility that similar peak shapes were obtained as a result of this bias needs to be discussed.

Technical corrections

Page 3, line 35: "may be associated"?

---

## Referee Comment (RC2) · Anonymous Referee #2 · 16 Jul 2020

The manuscript presented the chemical composition of individual sea salt aerosol particles (SSA) using a single particle aerosol mass spectrometer (SPAMS). About 50,000 SSA from 2 million detected particles were identified. These SSA were classified as SSA-aged, SSA-bio, and SSA-Ca. The manuscript showed positive correlations between Na and organic acids. Thus, the authors claimed that organic acids played significant role in the chloride depletion, then up to 34% of depletion was estimated. it also claims that SSA-bio particles can be assigned as the biological origin. This study provides additional data sets for the better understanding in the atmospheric processes of sea salt aerosol. Some of the conclusions need clarifications before it can be considered for publication.

Comments:

[Figure]

Line 22, 321-323. Please clarify which part of the SSA-bio particles are biological origin? Do you mean the SSA part in SSA-bio particles, like [Na] detected in the mass spec. are these components also biological origin? If so, is there any previous studies showing that biological sources produces SSA-like particles or components? If not, that means among SSA, half of them are sea salt aerosol (likely from sea spray) mixed with biological components during the transport, then current statement is misleading and not accurate.

Line 23, 36-41, 251-252, 308-309. It is not clear how these conclusions were reached. Line 251-252, it claims organic acids contributed about 2-34% chloride depletion in SSA-aged, and 2-39% in SSA-bio particles. Where do these numbers come from? Are these estimates based on SPAMS or bulk measurements as showing in Line 270-276? In Fig.S5, organic acids only contribute less the 20% of all acids. In Fig. 4 and Fig. 5, how can readers relate peak area of Cl/Na to the chloride depletion? For pure NaCl particles, what is the value of peak area of Cl/Na, if it is plotted in Fig.4/5?

Line 288-291, the relation/trend shown in Fig. 5 does not mean that they are "direct evidence".

Line 270-276, please provide details for the bulk measurements.

In the main text, please define "hourly mean peak area" in Figure 2 and "peak area of Cl/Na" (do you mean ratio of peak area of Cl to peak area of Na in y axis?) in Figure 4 and 5.

Figure 3, what are the standard deviations in these numbers?

---

## Author Comment (AC1) · 4 Aug 2020

**Point-by-point response to reviewer comment on manuscript acp-2020-443 "Impact of organic acids on chloride depletion of inland transported sea spray aerosols"**

We appreciate the two anonymous reviewers for their constructive comments to improve the manuscript. We have addressed every comment and made changes to improve the manuscript. With kindest regards, By Bojiang Su on behalf of all authors

**Note**

Referee Comments in black. Authors' Response in blue. Changes in manuscript in Red.

**Anonymous Referee #1**

**General comments**

This paper discusses the depletion of chloride in sea spray aerosol (SSA) particles based on a field observation of atmospheric aerosol during the East Asian summer monsoon using a single particle aerosol mass spectrometer. The authors claim that up to 34% of chloride depression could be explained by organic acids. They also claim that SSA of biogenic origin exhibits chloride depression that is distinctly different from that of commonly observed aged SSA, and infer that organic coatings on SSA could influence on the heterogeneous reactivity of SSA particles. The subject of this study is important to understand the chemical processes of SSA and also their role in the atmosphere.

Although the findings from this study are potentially significant, the conclusions are not fully supported by the experimental evidence provided in the current version of the manuscript. For example, in the quantitative analysis of the contributions of inorganic and organic acids on chloride depression, no explanation is provided about possible different sensitivity of the mass spectral analysis to different compounds. Further, the comparison of Cl/Na from mass spectra and that from mass concentrations was performed by omitting a substantial fraction of data points, without an adequate explanation on the rationale of the omission. To draw a solid conclusion from the data set presented in this paper, a thorough revision is necessary. More specific comments are listed below. **Author's Response:** We appreciate the valuable time and efforts from the referee to improve the manuscript. Please see below for the point-by-point response to reviewers' comments.

**R1-C1:** For example, in the quantitative analysis of the contributions of inorganic and organic acids on chloride depression, no explanation is provided about possible different sensitivity of the mass spectral analysis to different compounds.

Author's Response: Thanks for the referee's comment. In the revised manuscript, we have added a new section to discuss the quantitative analysis of single particle aerosol mass spectrometer. In addition, the possible different sensitivity of different species (especially Na, Cl, nitrate, sulfate and organics) in single particle mass spectrometry was also discussed.

"Using single particle mass spectrometry (SPMS) alone is subjected to the transmission efficiencies of particles through the aerodynamic lens, the possible selectivity and matrix effects on chemical components, resulting in inaccuracies of the number concentration, size distribution and chemical composition of the ambient aerosols (Gross et al., 2000; Qin et al., 2006; Pratt and Prather, 2012). Thus, a comparison analysis based on the obtained particle counts, size distributions and peak area / relative peak area via SPMAS in this study should be considered as the semi-quantitative analysis from statistical perspective (Hinz et al., 2005; Jeong et al., 2011; Healy et al., 2012; Zhou et al., 2016). More detailed discussion for the semi-quantitative analysis of SSA particles could be seen in *Supplement*."

**"1. Semi-quantitative analysis of SSA particles (in supplement)**

"Using single particle mass spectrometry (SPMS) alone is subjected to the transmission efficiencies of particles through the aerodynamic lens, the possible selectivity and matrix effects on chemical components, resulting in inaccuracies of the number concentration, size distribution and chemical composition of the ambient aerosols (Gross et al., 2000; Qin et al., 2006; Pratt and Prather, 2012).

Despite this, some species (e.g., sulfate, nitrate, sodium, ammonium, elemental carbon and organic carbon) presented relative strong correlations between peak area / relative peak area (data set from SPMS) and mass concentration (data set from other measurements such as micro-orifice uniform deposit impactor (MOUDI)), which indicated the real ambient condition could be reflected by SMPS in some extent (Qin et al., 2006; Jeong et al., 2011; Healy et al., 2012; Healy et al, 2013; Zhou et al., 2016). Besides, similar results of correlation analysis on Na+, Cl-, K+ for SSA particles (< 2  $\mu$ m) were also obtained, which suggested no significant additional composition effect for the desorption and ionization process (Dall'Osto et al., 2006).

In addition, for the core-shell structure of the most SSA particles (Collins et al., 2014; Chi et al., 2015), the pulse energy of the desorption and ionization laser via SPMS was influential for the selectivity of the detected components on the surface of the particles (Woods et al., 2002; Cai et al., 2006; Zelenyuk et al., 2008). Sultana et al. (2017) suggested that higher laser energy (> 1 mJ) could result in greater sodium signal, indicating more complete particle desorption and ionization. And lower laser energy (< 1 mJ) could be more likely to generate greater contribution to ion signature from the coating components such as organic species (Sultana et al., 2017). In order to reduce the impact of laser power on organic species, relative low laser energy ( $0.5 \pm 0.05$  mJ) was applied in this study.

Thus, a comparison analysis based on the obtained particle counts, size distributions and peak area / relative peak area via SPMAS in this study should be considered as the semi-quantitative analysis from statistical perspective (Hinz et al., 2005; Healy et al., 2012; Zhou et al., 2016)."

**R1-C2:** Further, the comparison of Cl/Na from mass spectra and that from mass concentrations was performed by omitting a substantial fraction of data points, without an adequate explanation on the rationale of the omission.

Author's Response: Thanks for the referee's comment. We have reviewed the original data set from the mass concentration and the mass spectral analysis. Only 3 data points should be excluded due to the maximum, minimum and the abnormal value. This part was reconsidered and rewritten in the revised manuscript as shown below:

"It is noted that excess  $[Cl]^-$  produced by fuel combustion (Lightowlers et al., 1988) could lower the %Cl- and high sensitivity of SPAMS to Na+ (Gross et al., 2000) could increase the Cl / Na value. These potential inaccuracies might be more likely to explain the weak difference for the two assessment methods of ageing degree of SSA particles, which are evaluated by hourly mean peak area of Cl / Na and quantification of chloride depletion on mass concentration. Generally, the quantitative analysis of mass concentration could directly evaluate the degree of chloride depletion in SSA particles (Braun et al., 2017). We note that there are some consistencies between the two assessment methods, which are supported by a positive correlation ( $r^2 = 0.22$ , p

On the other hand, Fig. S7 and Table S1 were modified and their description were also rewritten as shown below:

"Fig S7. Correlation analysis of the both assessment methods of chloride depletion. These data were conducted significance test and p < 0.001. The correlation analysis was carried out on 19 among 22 set of data due to the rest of the maximum, minimum and the abnormal value (detailed in Table S1)."

| Sampling set names  | Sampling dates | Duration (min) | Na + | Ca 2+ | Cľ   | NO 3 | SO4 2- | CI% depletion | Peak area ratio of Cl / Na |
|---------------------|----------------|----------------|-----------------|------------------|------|-----------------|-------------------|---------------|----------------------------|
| Cloud Water 1       | May 11         | 480            | 0.24            | 0.01             | 0.16 | 1.26            | 1.81              | 0.64          | 0.005*                     |
| Cloud Water 2       | May 26         | 150            | 1.31            | 0.74             | 0.58 | 5.59            | 9.32              | 0.65          | 0.023                      |
| Cloud Water 3       | May 26         | 140            | 1.73            | 0.98             | 0.56 | 3.90            | 6.48              | 0.74          | 0.028                      |
| Cloud Water 4       | May 26         | 180            | 5.24            | 1.80             | 0.99 | 6.41            | 9.65              | 0.87          | 0.025                      |
| Cloud Water 5       | May 27         | 214            | 1.98            | 1.21             | 1.08 | 9.33            | 13.84             | 0.55          | 0.029                      |
| Cloud Water 6       | May 30         | 270            | 1.85            | 1.35             | 0.53 | 8.57            | 13.08             | 0.73          | 0.030                      |
| Cloud Water 7       | May 30         | 305            | 1.88            | 1.07             | 0.79 | 11.14           | 16.95             | 0.67          | 0.028                      |
| Cloud Water 8       | May 30         | 515            | 1.68            | 1.22             | 0.83 | 11.59           | 19.32             | 0.55          | 0.026                      |
| Cloud Water 9       | Jun 1          | 225            | 1.07            | 0.79             | 0.31 | 5.57            | 5.71              | 0.73          | 0.028                      |
| Cloud Water 10      | Jun 2          | 230            | 0.99            | 0.42             | 0.05 | 2.10            | 3.84              | 0.96          | 0.011                      |
| Cloud Water 11      | Jun 2          | 141            | 1.63            | 0.59             | 0.30 | 3.84            | 5.64              | 0.88          | 0.016                      |
| Cloud Water 12      | Jun 2          | 134            | 1.44            | 0.46             | 0.15 | 1.85            | 2.63              | 0.93          | 0.014                      |
| Cloud Water 13      | Jun 2          | 203            | 3.79            | 1.03             | 0.35 | 3.85            | 5.80              | 0.94          | 0.046*                     |
| PM 2.5 1 | May 14-15      | 1440           | 1.12            | 1.04             | 0.38 | 1.05            | 2.60              | 0.62          | 0.038                      |
| PM 2.5 2 | May 18-19      | 1406           | 1.46            | 0.47             | 0.40 | 1.80            | 4.75              | 0.82          | 0.032                      |
| PM 2.5 3 | May 19-20      | 1423           | 0.45            | 0.22             | 0.07 | 2.07            | 5.09              | 0.88          | 0.031                      |
| PM 2.5 4 | May 20-21      | 1366           | 0.70            | 0.22             | 0.21 | 3.22            | 4.70              | 0.80          | 0.027                      |
| PM 2.5 5 | May 21-22      | 1431           | 0.48            | 0.18             | 0.13 | 2.11            | 3.52              | 0.82          | 0.029                      |
| PM 2.5 6 | May 22-23      | 1414           | 0.57            | 0.16             | 0.06 | 0.97            | 2.94              | 0.93          | 0.032                      |
| PM 2.5 7 | May 24-25      | 1419           | 0.72            | 0.21             | 0.17 | 2.59            | 4.22              | 0.80          | 0.054*                     |
| PM 2.5 8 | May 26-27      | 1391           | 0.72            | 0.47             | 0.25 | 2.36            | 3.20              | 0.70          | 0.029                      |
| PM 2.5 9 | May 29-30      | 1421           | 0.51            | 0.03             | 0.01 | 0.35            | 1.84              | 0.99          | 0.023                      |
| Mean                |                |                | 1.43            | 0.67             | 0.38 | 4.16            | 6.68              | 0.78          | 0.026                      |

\* These data are excluded because of the maximum (0.054), minimum (0.005) and abnomal value (0.046).

"Table S1. The major ions, chloride depletion (%Cl-) and hourly mean peak area ratio of Cl / Na in the sample of cloud water and PM2.5 during sampling period. The mass concentrations ( $\mu$ g m-3) of Na+, Ca2+, Cl-, NO3- and SO42- were analyzed using an ion chromatography (Metrohm, Herisau, Switzerland). The cloud water was sampled by a Caltech Active Strand Cloud Water Collector Version 2 (CASCC2) (Modini et al., 2015), when visibility was < 3 km until the volume exceeded 250 ml. The PM2.5 was sampled using an Atmospheric particle sampler (Mingye Environmental Protection Technology Co., Ltd., China) with an inlet cyclone with a cut-off aerodynamic diameter of 2.5 µm."

In **R1-C17**, we have discussed two assessment methods of chloride depletion (i.e., Cl / Na evaluated by hourly mean peak area and quantification on mass concentration.). Generally, the quantitative analysis of mass concentration could directly evaluate the degree of chloride depletion in SSA particles. We note that there are some consistencies between the two assessment methods, which are supported by a positive correlation ( $r^2 = 0.22$ , p < 0.001). Thus, the hourly mean peak area of Cl / Na could semi–quantitatively reflect the degree of chloride depletion, in some extent.

**Specific comments**

**R1-C3:** Page 3, lines 25-26: For nitrogen and sulfur, the explanation here sounds an overstatement. Author's Response: Thanks for pointing out this. The sentence was rewritten as "Heterogeneous reactions on sea spray aerosols (SSA) are the potential pathway to drive the circulation of chlorine, nitrogen, and sulfur in the atmosphere."

**R1-C4:** Page 6, line 80: Is there any experimental/theoretical evidence to support reaction R5? Author's Response: Exactly, Wang and Laskin, (2014) reported that internally mixed nitrate particles with organic acids could result in nitrate depletion and formation of organic salts, represented as:

NaNO3 (aq) + R-COOH (g or aq)  $\leftrightarrow$  HNO3 (g or aq) + R-COONa (s or aq)

**R1-C5:** Page 9, lines 146-151: If both types of particles were subjected to the analysis, it should be explained explicitly.

Author's Response: Thanks for the referee's opinion. We added a sentence:

"In the following discussion, the cloud residual particles and cloud-free particles were combined as the total of detected particles."

R1-C6: Page 10, line 163: What is the exact number of groups?

Author's Response: The detected particles were clustered into 8 groups by ART-2a. The sentence was rewritten as "A total of ~2 million detected particles were clustered into 8 groups using an Adaptive Resonance Theory neural network (ART-2a) (Song and Hopke, 1999) with a vigilance factor of 0.75, a learning rate of 0.05 and a maximum of 20 iterations."

**R1-C7:** Page 10, line 166: To justify the identification of SSA particles, a brief explanation on the differences between SSA particles and other types of particles should be provided.

Author's Response: We agree with the referee's opinion. And we added a sentence as below: "In single particle mass spectral fingerprints, it is the key to distinguish SSA from other groups of particles whether to present the ion signature of 23  $[Na]^+$ , 46  $[Na_2]^+$ , 62  $[Na_2O]^+$ , 63  $[Na_2OH]^+$ , 81  $[Na_2^{35}Cl]^+$  and 83  $[Na_2^{37}Cl]^+$  simultaneously. (Ault et al., 2014; Collins et al., 2014; Lin et al., 2019)."

**R1-C8:** Page 13, line 222: What can crystallize? The meaning of "precipitation" is not clear. Author's Response: Ault et al. (2014) and Sultana et al. (2017) observed the types of SSA particles associated with Na-deficient and Ca-rich mass spectral signatures. This Ca-rich population may be the coating of Na-rich particles (Sultana et al., 2017), which comes from the precipitation and crystallization by calcium sulfate, calcium sulfate hydrates, and sodium calcium sulfate salts during the dehydration of seawater and seawater droplets (Xiao et al., 2008).

The relevant sentences had been rewritten as below:

"This SSA population has been previously classified as "organic-carbon dominated (OC)" (Prather et al., 2013; Collins et al., 2014), likely resulted from the coating of Na-rich SSA particles through crystallization and precipitation of Ca-containing sulfate (e.g., calcium sulfate and sodium calcium sulfate) during the dehydration of sea water and seawater droplets (Xiao et al., 2008; Sultana et al., 2017; May et al., 2018)."

R1-C9: Page 13, lines 228-229: The definition of hourly-mean peak areas in Fig. 2 is unclear.

Author's Response: Thanks for pointing out this. We have given a definition of hourly mean peak area. This part has been rewritten as:

"The linear correlations based on hourly mean peak area between Na and chloride, sulfate, nitrate, and organic acids in the SSA-Aged and SSA-Bio particles are shown in Fig. 2. Herein, hourly mean peak area was defined as the mean peak area of a component for hourly detected particles, indicating the variations of chemical composition in individual particles."

**R1-C10:** Page 13, lines 232-233: This explanation is reasonable if nitrate was observed less frequently for non-SSA particles.

**Author's Response:** Thanks for the referee's opinion. It is true that relative low number fractions of nitrate (varying from 46% to 95%) were observed in other types of particles (i.e., Aged-EC, OC, K-rich and HM) at the same field site (Lin et al., 2017). However, for SSA particles, we emphasize the significance of heterogeneous reaction HNO3 (g or aq) + NaCl (s or aq)  $\rightarrow$  NaNO3 (aq) + HCl (g or aq) plays important role on chloride depletion of SSA particles during long-range transport.

**R1-C11:** Page 13, lines 236-238: It is difficult for readers to follow this part without an explanation about the mechanism of the suppression. Isn't it also possible that nitric acid and other acids have different reactivity to sea salts?

Author's Response: We agree with the referee's comment. For this part, we think that the relative high concentration of precursor  $NO_x$  was one of the factors in explanation of nitric acids responsible for chloride depletion.

The sentence had been rewritten as below:

"It is possible supported by the relative high concentration of its precursor  $NO_x$  (4.67 µg m-3) in the south China sector (Wang et al., 2016; Wu et al., 2019)."

**R1-C12:** Page 14, lines 245-246: The expression is misleading because the result in Fig. S4 is on number basis and may not relate to the mass concentrations of different organic salts in SSA particles.

Author's Response: We agree with the referee's comment. The relevant sentence had been rewritten as below:

"The detailed mixing state (by NFs) between SSA particles and several detected organic acids, as shown in Fig. S4, indicates that formate, oxalate, malonate, and glutarate might be the dominant organic salts. This result may be associated with the wide range of occurrence of corresponding organic acids in the atmosphere (Ghorai et al., 2014; Kawamura and Bikkina, 2016)."

**R1-C13:** Page 14, lines 246-258: In the absence of an appropriate explanation on possible different sensitivities of this mass spectrometry to different types of chemical components (sulfate/nitrate/organics), the discussion here is questionable.

Author's Response: Thanks for the referee's comment. We have added a detailed discussion on the possible different sensitivities and matrix effect of the mass spectrometry to different chemical component (including sulfate, nitrate, chloride, sodium and organics) in *Supplement* 1. Our obtained results were based on the semi-quantitative analysis from statistical perspective. In addition, we have referred the other citations as below:

"The similar method was also introduced to evaluate the extent of chloride depletion in previous study (Ault et al., 2014)."

R1-C14: Page 14, lines 258-261: Are the peak areas here hourly mean values?

Author's Response: It is correct. To make it clear, we have rewritten the sentence as below: "In addition, similar variations in hourly mean peak area of sulfate, nitrate and organic acids were observed in the SSA-Aged and SSA-Bio particles throughout the sampling period (Fig. S6), indicating a close connection of the formation mechanism between inorganic and organic acids."

**R1-C15:** Page 15, lines 264-265: Isn't there any possibility that matrix substances in the particles affect the relative sensitivity to detect Na and Cl?

Author's Response: Exactly, there is a possibility that matrix effect might be likely to affect the relative sensitivity to detect  $Na^+$  and  $Cl^-$ . The detailed discussion of this part was supplemented in *Supplement* 1.

**R1-C16:** Page 15, lines 272-275: The mass concentrations of Na+ and Cl- seem to be from ion chromatography but there is no explanation about the methodology in the main manuscript.

Author's Response: Thanks for the referee's opinion. We have supplemented the explanation of methodology related to ion chromatography in this part. As shown below:

"The mass concentration of major ions (such as  $Na^+$  and  $Cl^-$ ) was analyzed using an ion chromatography (Metrohm, Herisau, Switzerland), detailed information could be found in *supplement* Table S1. The overall %Cl- for the total SSA particles varying from 55% to 99% with an average of 78% (Table S1), which is similar to the previous field study in the PRD region reported by Chen et al. (2016)."

R1-C17: Page 15, line 280: Two assessment methods should be explained explicitly.

Author's Response: Thanks for the suggestion. We have rephrased this part to explain clearly.

"It is noted that excess  $[Cl]^-$  produced by fuel combustion (Lightowlers et al., 1988) could lower the %Cl- and high sensitivity of SPAMS to Na+ (Gross et al., 2000) could increase the Cl / Na value. These potential inaccuracies might be more likely to explain the weak difference for the two assessment methods of ageing degree of SSA particles, which are evaluated by hourly mean peak area of Cl / Na and quantification of chloride depletion on mass concentration. Generally, the quantitative analysis of mass concentration could directly evaluate the degree of chloride depletion in SSA particles (Braun et al., 2017). We note that there are some consistencies between the two assessment methods, which are supported by a positive correlation ( $r^2 = 0.22$ , p < 0.001) (Fig. S7). Hence, the hourly mean peak area of Cl / Na could also semi – quantitatively reflect the degree of chloride depletion, in some extent."

**R1-C18:** Page 15, line 281-283: The analysis of the correlation is problematic because 7 among 22 set of data were omitted from the analysis without an appropriate assessment.

Author's Response: Thanks for the referee's comment. We have reviewed the original data set from the mass concentration and the mass spectral analysis. Only 3 data points should be excluded due to the maximum, minimum and the abnormal value. Thus, the correlation analysis was carried out on 19 among 22 set of data. Detailed explanation could be found in **R1-C2**.

**R1-C19:** Page 33, line 623: Which part of data shows a normal distribution? Why it is obtained by the logarithmic transformation?

Author's Response: The data set related to hourly mean peak area of Na, nitrate, chloride and sulfate was logarithmically transformed to follow a normal distribution. From the statistical perspective, the correlation analysis should base on the data following the normal distribution. For the peak area of single particle mass spectrometry, there is a significant difference ( $1 \sim 4$  orders of magnitude) between maximum and minimum because of the different degree of desorption / ionization. In order to obtain the more accurate results, we thus used logarithmical transformation to make the data follow the normal distribution as possible.

**R1-C20:** Page 35, lines 632-635: An explanation on the box and whisker plots should be given **Author's Response:** Thanks for the referee's opinion. We supplemented this part as shown below:

"Figure 4. A box and whisker plot of hourly mean peak area of Cl / Na in SSA-Aged and SSA-Bio. Herein, the Cl / Na value is applied to evaluate the ageing degree of SSA particles. There is a significant difference of Cl / Na between the SSA-Aged and SSA-Bio (1.9% versus 5.4%, by mean value, respectively). In the box and whisker plot, the lower and upper lines of the box denote the 25 and 75 percentiles, respectively. The lower and upper edges denote the 10 and 90 percentiles, respectively."

**R1-C21:** Page 36, lines 637-639: Isn't it more reasonable to write Cl/Na as a function of peak areas of phosphate and organic nitrogen?

Author's Response: We agree with the referee's opinion. We have rewritten this sentence as below: "Figure 5. The hourly mean peak area ratio of Cl / Na varied as a function of logarithmical peak area of phosphate (m/z -63 and -79) and organic nitrogen (m/z -26 and -42)."

**R1-C22:** Page 2, lines 7-11 (supplement): The association between trajectories on the map and C1-C4 should be explained. The graph superimposed on the map should also be explained.

Author's Response: Thanks for the referee's opinion. We have rewritten and reorganized this part, as shown below:

"Figure S1. Quantitative distributions of SSA particles associated with clustered 72h back trajectories of air masses at 1800m above the ground during the sampling period (from 11 May to 3 June 2018). Four major cluster trajectories of air mass (namely C1, C2, C3, C4) were calculated by Meteoinfo (Wang, 2014) and plotted by Arcgis (Esri, Environmental Systems Research Institute, Inc.). Cluster 1 (41.64%) from Indo China Peninsula and Cluster 4 (18.46%) from south China Sea crossing through Hainan Peninsula carried the most (~ 25,000) and the least (~ 5,000) SSA particles, respectively. The both from the South China Sea Cluster 2 (28.92%) and Cluster 3 (10.98%), which brought approximately 12,500 and 7,500 SSA particles respectively, were rarely affected by anthropogenic emissions before reaching the coastline. Changes in proportions of the three types of SSA particles in different clusters were slight."

**R1-C23:** Page 4, lines 19-20 (supplement): If the detection efficiency for small particles was low, the peak shapes may be skewed. The possibility that similar peak shapes were obtained as a result of this bias needs to be discussed.

**Author's Response:** We agree with the referee's comment. There is a possibility that the peak shapes may be skewed due to the low detection efficiency for small particles ( $d_{va}

Figure R1. The simulational and experimental transmission efficiency of aerodynamic lens in SPAMS.

**Technical corrections**

**R1-C24:** Page 3, line 35: "may be associated"? Author's Response: Thanks for the technical suggestions. Revisions has been made accordingly.

**Anonymous Referee #2**

**General comments**

The manuscript presented the chemical composition of individual sea salt aerosol particles (SSA) using a single particle aerosol mass spectrometer (SPAMS). About 50,000 SSA from 2 million detected particles were identified. These SSA were classified as SSA-aged, SSA-bio, and SSA-Ca. The manuscript showed positive correlations between Na and organic acids. Thus, the authors claimed that organic acids played significant role in the chloride depletion, then up to 34% of depletion was estimated. It also claims that SSA-bio particles can be assigned as the biological origin. This study provides additional data sets for the better understanding in the atmospheric processes of sea salt aerosol. Some of the conclusions need clarifications before it can be considered for publication.

Author's Response: We appreciate the valuable time and efforts from the referee to improve the manuscript. Please see below for the point-by-point response to reviewers' comments.

**Specific comments**

**R2-C1:** Line 22, 321-323. Please clarify which part of the SSA-bio particles are biological origin? Do you mean the SSA part in SSA-bio particles, like [Na] detected in the mass spec. are these components also biological origin? If so, is there any previous studies showing that biological sources produces SSA-like particles or components? If not, that means among SSA, half of them are sea salt aerosol (likely from sea spray) mixed with biological components during the transport, then current statement is misleading and not accurate.

**Author's Response:** We agree with the reviewer's opinion that current statement is misleading and not accurate. In this study, three types of SSA particles with distinct mass spectral characteristics were obtained by using an Adaptive Resonance Theory neural network (ART-2a), including SSA-Aged, SSA-Bio and SSA-Ca. For SSA-Bio, it was identified not only by the general characteristics of the SSA (i.e., 23 [Na]+, 46 [Na2]+, 62 [Na2O]+, 63 [Na2OH]+, 81 [Na235Cl]+ and 83 [Na237Cl]+), but also identified by the significant additional organic ion markers (e.g., amine (m/z: 58 [C2H5NHCH2]+ and 59 [N(CH3)3]+), organic nitrogen (m/z: -26 [CN]- or 42 [CNO]-) and phosphate (m/z: -63 [PO2]-, 79 [PO3]-)). This population of SSA with the above mentioned organic matters was identified as the "BioSS" in previous laboratory studies (Ault et al., 2014; Sultana et al., 2017). It is more likely produced by the ejection of SSA with microbes (e.g., phytoplankton and bacteria) in the sea surface microlayer (Cochran et al., 2017a; Cochran et al., 2017b; Wang et al., 2017). Thus, the mass spectra of SSA-Bio may be likely produced by the microbial, microbe-containing, or microbe-fragment-containing SSA particles, which is also similar with previous studies (Steele et al., 2003; Czerwieniec et al., 2005; Srivastava et al., 2005; Sultana et al., 2017).

We have rephrased this sentences as the summary of SSA-Bio particles as below:

"Ault et al. (2014) and Sultana et al. (2017) have also observed a special SSA type with higher biological ion markers in laboratory studies, which indicates it is more likely produced by the ejection of SSA with microbes (e.g., phytoplankton and bacteria) in SSML (Patterson et al., 2016; Cochran et al., 2017a; Cochran et al., 2017b; Wang et al., 2017)."

R2-C2: Line 23, 36-41, 251-252, 308-309. It is not clear how these conclusions were reached.

**Author's Response:**

Line 23: Organic acids considerably contribute to chloride depletion of SSA particles.

**The** results of strong correlation on hourly mean peak area between Na and organic acids and high number fraction of organic acids in SSA indicate the possible presence of organic salts in SSA and the substantial contribution of organic acids to the chloride depletion (Fig. 2 and Fig. 3). Additionally, the evaluation by relative peak area ratio also shows organic acids considerably contribute to chloride depletion of SSA particles.

Line 36-41: Strongly positive correlations between Na and organic acids (including formate, acetate, propionate, pyruvate, oxalate, malonate, succinate, and glutarate) were observed for the SSA-Aged ( $r^2 = 0.52$ , p < 0.01) and SSA-Bio ( $r^2 = 0.61$ , p < 0.01), indicating the significance of organic acids in the chloride depletion during inland transport.

**In** single particle mass spectrometry, the variation of peak area in time series could reveal the reaction mechanism of atmospheric chemistry (Zauscher et al., 2013). The correlation analysis based on peak area of different species could reflect the chemical composition of some types of particles in some content (Peng et al., 2019). Thus, we think that a perspective on correlation analysis between Na and organic acids may be likely to reflect the organic acids to chloride depletion.

**Line 249-252:** In the ageing process of the SSA particles, nitrate occupies a large proportion in the SSA-Aged (63-96%) and SSA-Bio particles (64-95%), respectively. Notably, chloride depletion attributed to organic acids could account for 2–34% in the SSA-Aged particles and 2–29% in the SSA-Bio particles.

**The** hourly mean relative peak area (RPA) ratio (acids / (sulfate + nitrate + organic acids)) is further applied to roughly evaluate the relative contribution of different acids (nitric acid, sulfuric acid, and organic acids) to the chloride depletion of SSA particles. The similar method was also introduced to evaluate the extent of chloride depletion in previous study (Ault et al., 2014). Further explanation cloud be seen in **R2-C3**.

Line 308-309: Up to 34% of chloride depletion could be explained by diverse organic acids. The evaluation of acids to chloride depletion could be seen in **R2-C3**. We have reploted the Figure. S5 to exhibit the scale range of the relative contribution of acids to the chloride depletion in SSA particles. Based on this method, we think that up to 34% of chloride depletion could be explained by diverse organic acids.

**R2-C3:** Line 251-252, it claims organic acids contributed about 2-34% chloride depletion in SSA-aged, and 2-39% in SSA-bio particles. Where do these numbers come from? Are these estimates based on SPAMS or bulk measurements as showing in Line 270-276?

Author's Response: We used the data set via SPAMS of hourly mean relative peak area (RPA) ratio (i.e., acids / (sulfate + nitrate + organic acids)) to roughly evaluate the relative contribution of different acids to the chloride depletion of SSA particles. Each of hourly mean RPA ratio makes up the scale range of the relative contribution of acids to the chloride depletion in SSA particles. In order to further improve the reliability of this evaluation, we have referred the other citations as below:

"The similar method was also introduced to evaluate the extent of chloride depletion in previous study (Ault et al., 2014)."

**R2-C4:** In Fig.S5, organic acids only contribute less the 20% of all acids.

Author's Response: Thanks for the reviewer's opinion. We have also noticed the possibility of misleading from this part. We have reploted the Figure. S5 to exhibit the scale range of the relative contribution of acids to the chloride depletion in SSA particles.

Figure S5. The relative contribution of different acids (nitric acid, sulfuric acid, and organic acids) to the chloride depletion of SSA particles. The ratio referred to the hourly mean relative peak area (RPA) ratio (acids / (sulfate + nitrate + organic acids)). Sulfate, nitrate and organic acids referred to peaks at m/z -97, m/z -46 and -62, and the mentioned organic acids in Fig. S4, respectively.

**R2-C5:** In Fig. 4 and Fig. 5, how can readers relate peak area of Cl/Na to the chloride depletion? For pure NaCl particles, what is the value of peak area of Cl/Na, if it is plotted in Fig.4/5?

Author's Response: Thanks for the reviewer's opinion. In order to improve the understanding for the broader readers, we have rephrased the figure caption of Figure. 4 and given the definition for Cl / Na in main text.

"Figure 4. A box and whisker plot of hourly mean peak area of Cl / Na in SSA-Aged and SSA-Bio. Herein, the Cl / Na value is applied to evaluate the ageing degree of SSA particles. There is a significant difference of Cl / Na between the SSA-Aged and SSA-Bio (1.9% versus 5.4%, by mean value, respectively). In the box and whisker plot, the lower and upper lines of the box denote the 25 and 75 percentiles, respectively. The lower and upper edges denote the 10 and 90 percentiles, respectively."

The definition of Cl/ Na is given in main text:

"In this study, Cl / Na value was defined as the ratio of hourly mean peak area of Cl to Na, which presented the degree of the chloride depletion in SSA particles."

For pure NaCl particles, we could hardly obtain the mass spectra because of the few responses of NaCl particles to 266 nm Nd:YAG laser with ca. 0.5 mJ laser energy. Briefly, there is a challenge for ionization and desorption of pure NaCl particles with the crystalline structure because of the disparity between the first ionization potentials of sodium (5.1 eV) and chlorine (13 eV) (Reents and Ge, 2000; Zawadowicz et al., 2015). If both easily ionized and difficult ionized species are simultaneously present, then charge will be transferred within the ion cloud from high-ionization-potential cations to neutral species having a low-ionization-potential (Reents and Schabel, 2001; Zawadowicz et al., 2015). Thus this, it is difficult to produce surface ionization for pure NaCl particles due to the relative high-ionization-potentials (Sinha and Friedlander, 1985, Hatch et al., 2011).

**R2-C6:** Line 288-291, the relation/trend shown in Fig. 5 does not mean that they are "direct evidence".

Author's Response: Thanks for the reviewer's opinion. We have deleted the "direct".

| Sampling set names  | Sampling dates | Duration (min) | Na + | Ca 2+ | Cľ   | NO 3 | SO4 2- | Cl ® % depletion | Peak area ratio of Cl / Na |
|---------------------|----------------|----------------|-----------------|------------------|------|-----------------|-------------------|-----------------------------|----------------------------|
| Cloud Water 1       | May 11         | 480            | 0.24            | 0.01             | 0.16 | 1.26            | 1.81              | 0.64                        | 0.005*                     |
| Cloud Water 2       | May 26         | 150            | 1.31            | 0.74             | 0.58 | 5.59            | 9.32              | 0.65                        | 0.023                      |
| Cloud Water 3       | May 26         | 140            | 1.73            | 0.98             | 0.56 | 3.90            | 6.48              | 0.74                        | 0.028                      |
| Cloud Water 4       | May 26         | 180            | 5.24            | 1.80             | 0.99 | 6.41            | 9.65              | 0.87                        | 0.025                      |
| Cloud Water 5       | May 27         | 214            | 1.98            | 1.21             | 1.08 | 9.33            | 13.84             | 0.55                        | 0.029                      |
| Cloud Water 6       | May 30         | 270            | 1.85            | 1.35             | 0.53 | 8.57            | 13.08             | 0.73                        | 0.030                      |
| Cloud Water 7       | May 30         | 305            | 1.88            | 1.07             | 0.79 | 11.14           | 16.95             | 0.67                        | 0.028                      |
| Cloud Water 8       | May 30         | 515            | 1.68            | 1.22             | 0.83 | 11.59           | 19.32             | 0.55                        | 0.026                      |
| Cloud Water 9       | Jun 1          | 225            | 1.07            | 0.79             | 0.31 | 5.57            | 5.71              | 0.73                        | 0.028                      |
| Cloud Water 10      | Jun 2          | 230            | 0.99            | 0.42             | 0.05 | 2.10            | 3.84              | 0.96                        | 0.011                      |
| Cloud Water 11      | Jun 2          | 141            | 1.63            | 0.59             | 0.30 | 3.84            | 5.64              | 0.88                        | 0.016                      |
| Cloud Water 12      | Jun 2          | 134            | 1.44            | 0.46             | 0.15 | 1.85            | 2.63              | 0.93                        | 0.014                      |
| Cloud Water 13      | Jun 2          | 203            | 3.79            | 1.03             | 0.35 | 3.85            | 5.80              | 0.94                        | 0.046*                     |
| PM 2.5 1 | May 14-15      | 1440           | 1.12            | 1.04             | 0.38 | 1.05            | 2.60              | 0.62                        | 0.038                      |
| PM 2.5 2 | May 18-19      | 1406           | 1.46            | 0.47             | 0.40 | 1.80            | 4.75              | 0.82                        | 0.032                      |
| PM 2.5 3 | May 19-20      | 1423           | 0.45            | 0.22             | 0.07 | 2.07            | 5.09              | 0.88                        | 0.031                      |
| PM 2.5 4 | May 20-21      | 1366           | 0.70            | 0.22             | 0.21 | 3.22            | 4.70              | 0.80                        | 0.027                      |
| PM 2.5 5 | May 21-22      | 1431           | 0.48            | 0.18             | 0.13 | 2.11            | 3.52              | 0.82                        | 0.029                      |
| PM 2.5 6 | May 22-23      | 1414           | 0.57            | 0.16             | 0.06 | 0.97            | 2.94              | 0.93                        | 0.032                      |
| PM 2.5 7 | May 24-25      | 1419           | 0.72            | 0.21             | 0.17 | 2.59            | 4.22              | 0.80                        | 0.054*                     |
| PM 2.5 8 | May 26-27      | 1391           | 0.72            | 0.47             | 0.25 | 2.36            | 3.20              | 0.70                        | 0.029                      |
| PM 2.5 9 | May 29-30      | 1421           | 0.51            | 0.03             | 0.01 | 0.35            | 1.84              | 0.99                        | 0.023                      |
| Mean                |                |                | 1.43            | 0.67             | 0.38 | 4.16            | 6.68              | 0.78                        | 0.026                      |

| R2-C7: Line 270-276 | please provide details for the bulk measurements.              |       |
|----------------------------|----------------------------------------------------------------|-------|
| Author's Response:         | Details for the bulk measurement and SPAMS analysis as shown b | elow: |

\* These data are excluded because of the maximum (0.054), minimum (0.005) and abnomal value (0.046).

Table S1. The major ions, chloride depletion (%Cl-) and hourly mean peak area ratio Cl / Na in the sample of cloud water and PM2.5 during sampling period. The mass concentrations ( $\mu$ g m-3) of Na+, Ca2+, Cl-, NO3- and SO42- were analyzed using an ion chromatography (Metrohm, Herisau, Switzerland). The cloud water was sampled by a Caltech Active Strand Cloud Water Collector Version 2 (CASCC2) (Modini et al., 2015), when visibility was < 3 km until the volume exceeded 250 ml. The PM2.5 was sampled using an Atmospheric particle sampler (Mingye Environmental Protection Technology Co., Ltd., China) with an inlet cyclone with a cut-off aerodynamic diameter of 2.5 µm.

In table S1, we simultaneously obtained the %Cl- depletion and hourly mean peak area ratio Cl / Na corresponding to the sampling period. Then, we compared the degree of chloride depletion of SSA particles in quantitative analysis by mass concentration and semi-quantitative analysis by hourly mean peak area ratio.

Fig S7. Correlation analysis of the both assessment methods of chloride depletion. These data were conducted significance test and p < 0.001. The correlation analysis was carried out on 19 among 22 set of data due to the rest of the maximum, minimum and the abnormal value (detailed in Table S1).

**R2-C8:** In the main text, please define "hourly mean peak area" in Figure 2 and "peak area of Cl/Na" (do you mean ratio of peak area of Cl to peak area of Na in y axis?) in Figure 4 and 5. **Author's Response:** Thanks for the reviewer's opinion. We have given a definition for that.

"The linear correlations based on hourly mean peak area between Na and chloride, sulfate, nitrate, and organic acids in the SSA-Aged and SSA-Bio particles are shown in Fig. 2. Herein, hourly mean peak area was defined as the mean peak area of a component for hourly detected particles, indicating the variations of chemical composition in individual particles."

"In this study, Cl / Na value was defined as the ratio of hourly mean peak area of Cl to Na, which presented the degree of the chloride depletion in SSA particles."

**R2-C9: Figure 3, what are the standard deviations in these numbers?**

Author's Response: The number of individual particles was assumed to follow the Poisson distribution (Pratt et al., 2010; Lin et al., 2017), standard deviations for the hourly mean number fractions of major component in SSA-Aged and SSA-Bio as shown below:

| Number fractions    | Chloride | Sulfate | Organic acids | Nitrate | Ammonium | Amine  | CN/CNO | Phosphate | $C_2H_3$ | $C_2H_3O$                       |
|---------------------|----------|---------|---------------|---------|----------|--------|--------|-----------|----------|---------------------------------|
| SSA-Aged            | 0.50     | 0.30    | 0.72          | 0.99    | 0.09     | 0.11   | 0.56   | 0.22      | 0.08     | 0.04                            |
| SSA-Bio             | 0.36     | 0.62    | 0.59          | 1.00    | 0.34     | 0.62   | 0.54   | 0.60      | 0.16     | 0.19                            |
|                     |          |         |               |         |          |        |        |           |          |                                 |
| Standard deviations | Chloride | Sulfate | Organic acids | Nitrate | Ammonium | Amine  | CN/CNO | Phosphate | $C_2H_3$ | C 2 H 3 O |
| SSA-Aged            | 0.0045   | 0.0035  | 0.0054        | 0.0063  | 0.0019   | 0.0022 | 0.0048 | 0.0031    | 0.0018   | 0.0014                          |
| SSA-Bio             | 0.0039   | 0.0051  | 0.0050        | 0.0064  | 0.0038   | 0.0051 | 0.0047 | 0.0050    | 0.0026   | 0.0029                          |

**References**

- Ault, A. P., Guasco, T. L., Baltrusaitis, J., Ryder, O. S., Trueblood, J. V., Collins, D. B., Ruppel, M. J., Cuadra-Rodriguez, L. A., Prather, K. A., and Grassian, V. H.: Heterogeneous reactivity of nitric acid with nascent sea spray aerosol: Large differences observed between and within individual particles, J. Phys. Chem. Lett., 5, 2493-2500, http://doi.org/10.1021/jz5008802, 2014.
- Braun, R. A., Dadashazar, H., MacDonald, A. B., Aldhaif, A. M., Maudlin, L. C., Crosbie, E., Aghdam, M. A., Hossein Mardi, A., and Sorooshian, A.: Impact of Wildfire Emissions on Chloride and Bromide Depletion in Marine Aerosol Particles, Environ. Sci. Technol., 51, 9013-9021, http://doi.org/10.1021/acs.est.7b02039, 2017.
- Cai, Y., Zelenyuk., A., and Imre., D.: A High Resolution Study of the Effect of Morphology On the Mass Spectra of Single PSL Particles with Na-containing Layers and Nodules, Aerosol. Sci. Tech, 40, 1111–1122, http://doi.org/10.1080/02786820601001677, 2006.
- Chi, J. W., Li, W. J., Zhang, D. Z., Zhang, J. C., Lin, Y. T., Shen, X. J., Sun, J. Y., Chen, J. M., Zhang, X. Y., Zhang, Y. M., and Wang, W. X.: Sea salt aerosols as a reactive surface for inorganic and organic acidic gases in the Arctic troposphere, Atmos. Chem. Phys., 15, 11341-11353, http://doi.org/10.5194/acp-15-11341-2015, 2015.
- Chen, W., Wang, X., Cohen, J. B., Zhou, S., Zhang, Z., Chang, M., and Chan, C.-Y.: Properties of aerosols and formation mechanisms over southern China during the monsoon season, Atmos. Chem. Phys., 16, 13271-13289, http://doi.org/10.5194/acp-16-13271-2016, 2016.
- Cochran, R. E., Laskina, O., Trueblood, J. V., Estillore, A. D., Morris, H. S., Jayarathne, T., Sultana, C. M., Lee, C., Lin, P., Laskin, J., Laskin, A., Dowling, J. A., Qin, Z., Cappa, C. D., Bertram, T. H., Tivanski, A. V., Stone, E. A., Prather, K. A., and Grassian, V. H.: Molecular Diversity of Sea Spray Aerosol Particles: Impact of Ocean Biology on Particle Composition and Hygroscopicity, Chem., 2, 655-667, http://doi.org/10.1016/j.chempr.2017.03.007, 2017a.

- Cochran, R. E., Ryder, O. S., Grassian, V. H., and Prather, K. A.: Sea Spray Aerosol: The Chemical Link between the Oceans, Atmosphere, and Climate, Accounts Chem Res, 50, 599-604, http://doi.org/10.1021/acs.accounts.6b00603, 2017b.
- Collins, D. B., Zhao, D. F., Ruppel, M. J., Laskina, O., Grandquist, J. R., Modini, R. L., Stokes, M. D., Russell, L. M., Bertram, T. H., Grassian, V. H., Deane, G. B., and Prather, K. A.: Direct aerosol chemical composition measurements to evaluate the physicochemical differences between controlled sea spray aerosol generation schemes, Atmos. Meas. Tech., 7, 3667-3683, http://doi.org/10.5194/amt-7-3667-2014, 2014.
- Czerwieniec, G. A., Russell., S. C., Tobias., H. J., Pitesky., M. E., Fergenson., D. P., Steele., P., Srivastava., A., Horn., J. M., Frank., M., Gard., E. E., and Lebrilla., C. B.: Stable Isotope Labeling of Entire Bacillus atrophaeus Spores and Vegetative Cells Using Bioaerosol Mass Spectrometry, Anal. Chem., 77, 1081-1087, http://doi.org/10.1021/ac0488098, 2005.
- Dall'Osto, M., Harrison., R. M., and Beddows., D. c. S.: Single-Particle Detection Efficiencies of Aerosol Time-of-Flight Mass Spectrometry during the North Atlantic Marine Boundary Layer Experiment, Environ. Sci. Technol., 40, 5029-5035, http://doi.org/10.1021/es050951i, 2006.
- Ghorai, S., Wang, B., Tivanski, A., and Laskin, A.: Hygroscopic properties of internally mixed particles composed of NaCl and water-soluble organic acids, Environ. Sci. Technol., 48, 2234-2241, http://doi.org/10.1021/es404727u, 2014.
- Gross, D. S., Galli., M. E., Silva., P. J., and Prather., a. K. A.: Relative Sensitivity Factors for Alkali Metal and Ammonium Cations in Single-Particle Aerosol Time-of-Flight Mass Spectra, Anal. Chem., 72, 416-422, http://doi.org/10.1021/ac990434g, 2000.
- Hatch, L. E., Creamean, J. M., Ault, A. P., Surratt, J. D., Chan, M. N., Seinfeld, J. H., Edgerton, E. S., Su, Y. X., and Prather, K. A.: Measurements of Isoprene-Derived Organosulfates in Ambient Aerosols by Aerosol Time-of-Flight Mass Spectrometry Part 1: Single Particle Atmospheric Observations in Atlanta, Environ. Sci. Technol., 45, 5105-5111, http://doi.org/10.1021/es103944a, 2011.
- Healy, R. M., Sciare, J., Poulain, L., Kamili, K., Merkel, M., Müller, T., Wiedensohler, A., Eckhardt, S., Stohl, A., Sarda-Estève, R., McGillicuddy, E., amp, apos, Connor, I. P., Sodeau, J. R., and Wenger, J. C.: Sources and mixing state of size-resolved elemental carbon particles in a European megacity: Paris, Atmos. Chem. Phys., 12, 1681-1700, http://doi.org/10.5194/acp-12-1681-2012, 2012.
- Healy, R. M., Sciare, J., Poulain, L., Crippa, M., Wiedensohler, A., Prévôt, A. S. H., Baltensperger, U., Sarda-Estève, R., McGuire, M. L., Jeong, C. H., McGillicuddy, E., amp, apos, Connor, I. P., Sodeau, J. R., Evans, G. J., and Wenger, J. C.: Quantitative determination of carbonaceous particle mixing state in Paris using single-particle mass spectrometer and aerosol mass spectrometer measurements, Atmos. Chem. Phys., 13, 9479-9496, http://doi.org/10.5194/acp-13-9479-2013, 2013.
- Hinz, K.-P., Trimborn, A., Weingartner, E., Henning, S., Baltensperger, U., and Spengler, B.: Aerosol single particle composition at the Jungfraujoch, J. Aerosol. Sci, 36, 123-145, http://doi.org/10.1016/j.jaerosci.2004.08.001, 2005.
- Jeong, C. H., McGuire, M. L., Godri, K. J., Slowik, J. G., Rehbein, P. J. G., and Evans, G. J.: Quantification of aerosol chemical composition using continuous single particle measurements, Atmos. Chem. Phys., 11, 7027-7044, http://doi.org/10.5194/acp-11-7027-2011, 2011.
- Kawamura, K., and Bikkina, S.: A review of dicarboxylic acids and related compounds in atmospheric

aerosols: Molecular distributions, sources and transformation, Atmos. Res., 170, 140-160, http://doi.org/10.1016/j.atmosres.2015.11.018, 2016.

- Lightowlers, P. J., and Cape, J. N.: Sources and fate of atmospheric HCl in the U.K. and western Europe, Atmos. Environ., 22, 7-15, http://doi.org/10.1016/0004-6981(88)90294-6, 1988.
- Lin, Q, Yang, Y., Fu, Y., Zhang, G., Jiang, F.,